# Effect of Welding Defects on Fatigue Properties of SWA490BW Steel Cruciform Welded Joints

**DOI:** 10.3390/ma16134751

**Published:** 2023-06-30

**Authors:** Xingyuan Xu, Liyang Xie, Song Zhou, Jinlan An, Yanqing Huang, Youcheng Liu, Lei Jin

**Affiliations:** 1School of Mechanical Engineering and Automation, Northeastern University, Shenyang 110004, China; 15998808199@163.com (X.X.); lyxie@mail.neu.edu.cn (L.X.); 2Key Laboratory of Fundamental Science for National Defense of Aeronautical Digital Manufacturing Process, Shenyang Aerospace University, Shenyang 110136, China; dl23anjinlan@163.com (J.A.); hyq024@126.com (Y.H.); 3School of Mechanical and Electrical Engineering, Shenyang Aerospace University, Shenyang 110006, China; liuyouc0606@163.com (Y.L.); jinlei202303@163.com (L.J.)

**Keywords:** welded joints, defect, fatigue limit, stress concentration factor

## Abstract

Welding is prone to several defects. To test the fatigue properties of the welded defective joints of high-speed rail bogies, SMA490BW steel cruciform welded joints were employed with artificial defects treatment. Consequently, fatigue tests were conducted on the specimens. Fatigue fracture morphology was studied via scanning electron microscopy. The ABAQUS (version 2022) finite element software was used to calculate the stress distribution and concentration factor of cruciform welded joints with defects. The results show that the fatigue limits of 1 and 2.4 mm defect specimens were approximately 57.2 and 53.75 Mpa, respectively. Furthermore, the stress concentration factor of no, 1 mm, and 2.4 mm defects were 2.246, 4.441, and 6.684, respectively, indicating that the stress concentration factor of 1 and 2.4 mm defects increased by 98 and 198%, respectively, with respect to the no-defect case.

## 1. Introduction

At present, the high-speed rail in China is developing rapidly, and the primary function of high-speed rail bogies is to carry mass; thus, the quality of high-speed rail bogies is related to the safety of trains and passengers [1]. Welding is a basic technology for manufacturing high-speed rail bogies. Consequently, welding quality is an important symbol of high-speed rail bogie quality, and the welding efficiency directly affects the process and cost of manufacturing bogies [2]. Welding is often used to join individual parts of a bogie during its manufacturing process. As the bogies will bear considerable loads during the operation of the high-speed rail, they must have reliable welding quality at the junction of the parts to ensure the safe operation of the high-speed rail. The probability of fatigue failure can reach up to 70–90% in rail transport [3]. Thus, studies on the quality of the welded joints of bogies are necessary. It is of great significance to ensure the safety and reliability of the welded joints of bogies during the working process. The S-N curve is among the main methods for examining welded joints in terms of fatigue failure. SMA490BW steel exhibits good welding performance, high strength, and reliability; thus, it is widely used on domestic high-speed rails. The bogies of the currently used CRH-2 high-speed rail train are welded and manufactured using SMA490BW steel.

The fatigue properties of SMA490BW steel-welded joints have been extensively researched both at home and abroad. In China, Xu Liang et al. studied the fatigue crack propagation behavior and fatigue properties of notched specimens of SMA490BW steel [4,5,6]. Wang Lei et al. studied the effect of grinding treatment method on the fatigue properties of cruciform welded joints of SMA490BW steel [7]. Xu et al. and Kunqiang et al. studied the welding method and corrosion resistance of SMA490BW steel [8,9]. Furthermore, He Belin et al. studied the effects of ultrasonic impact and mechanical polishing on the fatigue properties of SMA490BW steel. Additionally, they examined very high cycle fatigue properties of welded joints [10,11,12,13,14]. Yu et al. studied the effects of different post-weld heat treatments on the mechanical properties and residual stress of SMA490BW steel [15]. Liu et al. studied the effect of work-piece grooves on the mechanical properties of SMA490BW and 304 stainless steel dissimilar joints [16]. Wang et al. discussed the causes of crack formation in the weld toe of SMA490BW steel bogies [17].

The above conclusions were obtained through extensive studies of a complete specimen; however, according to the construction manual, welding can be defective in joints. In other words, there may be welded defects that have not yet reached the defect state and, thus, can still be used normally. These defects have specific size requirements. However, no study has been conducted despite the obvious existence of such defective welded joints. The lifespan of the steering bogie is generally not less than 3 million kilometers, but defects in the structure will inevitably occur with the operation of the steering bogie. In this study, we used the materials and welding technology of the steering bogie to obtain a cross-welded joint consistent with the one on the steering bogie and then artificially introduced defects into the specimens to study the fatigue life of cross-welded joints with different defect sizes. We investigated the effects of cracks on the strength characteristics of the cross-welded joints in the steering bogie. Through this study, we can gain an understanding of the impact of defects on the welded joint.

## 2. Test Materials and Methods

The test material selected was flow-alloy weathering steel SMA490BW plate, and its chemical composition is presented in Table 1. The conventional mechanical property test results are presented in Table 2. The original steel plate size used for welded joints measured 300 mm × 150 mm × 12 mm. Furthermore, the MAG melting welding method was used to fabricate the welded joints, with a shielding gas of 80% Ar + 20% CO_2_ and CHW55-CNH welded wire with a diameter of 1.6 mm. After welding was completed, sampling was carried out, and the original fatigue test specimen is shown in Figure 1. The overall fatigue test specimens of the welded joint were conducted in accordance with the method of axial loading fatigue test of metal materials specified in HB5287-1996.

The original specimen was treated with defects, which refers to the use of wire cutting at the welding point of the cruciform welded joints to cut into a certain depth of defects. In this experiment, the crack sizes were 1 and 2.4 mm. According to the construction manual, the welding allowed incomplete penetration of 1 mm, and when the full penetration depth reached 80% of the plate thickness in ultrasonic inspection, the unmelted size was 2.4 mm. The exact location of the defect is shown in Figure 2.

Using a QBG-100 high-frequency fatigue testing machine, we employed the GB3075-82 “Metal Axial Fatigue Test Method” to perform the fatigue test. The test conditions were room temperature air and stress ratio R = −1 with sine-wave loading. Merlin Compact-type scanning electron microscopy was used to observe the fracture of SMA490BW weathering steel cruciform welded joints.

### 2.1. Effect of Defect Size on Fatigue Properties of Welded Joints

The specific results of the fatigue test on no, 1 mm, and 2.4 mm defects treatment are presented in Table 3, Table 4 and Table 5. Figure 3 shows the fatigue test results. The test data were 90% confidence under 50% survival rate, and the S-N curve of SMA490BW steel under stress ratio R = −1 was fitted by the least squares method. Table 6 presents the S-N curvilinear relationship equation corresponding to different defect sizes. The values in parentheses indicate that the specimen is broken at the weld toe, and the bracketed values indicate that the specimen is broken at the weld in Table 3, Table 4 and Table 5.

From Figure 3, at R = −1, the defect-free curve was at the top, followed by the curve with a defect size of 1 mm, with that of 2.4 mm at the bottom. The fatigue limits of the 1 and 2.4 mm defect specimens were 57.2 and 53.7 Mpa, respectively. Thus, the fatigue limit of the 1 mm defect specimen was 6.1% higher than that of 2.4 mm defect specimen. However, compared with the 70.9 Mpa welded non-defective specimen, the fatigue limits of the 1 and 2.4 mm defect specimens were reduced by 18.9% and 23.9%, respectively. This shows that within 80% of the minimum penetration, different defect sizes have minimal effect on the fatigue limit of welding; however, the existence of defects significantly reduces the fatigue limit of the specimen.

### 2.2. Morphology of Fatigue Fractures

Defects are caused by wire cutting at the weld of the specimen, and the appearance of defects results in stress concentration, which will result in that area being more prone to breakage than the other areas. In this test, when the defect was 1 mm, 13 of the total 25 test pieces broke at the defect, accounting for 52% of the specimens. When the defect was 2.4 mm, however, 17 of the 25 test pieces broke at the defect, accounting for 68%. The rest of the test pieces broke at the weld toes. This is attributed to the effect of welding, which results in the stress concentration in the welded toe area [18,19,20,21,22]. Figure 4 and Figure 5 show the fracture diagrams at the defect and weld toe, respectively. Fewer cracks were initiated at the defects, and they were roughly perpendicular to the notch caused by wire cutting. Owing to being at the edge of wire cutting, the stress value here was roughly the same, and the stress at the notch caused by wire cutting at this time was the highest. However, several cracks were initiated at the fracture on the weld toe. They exhibited a curvilinear form, which was caused by stress magnitude in different area. In the fatigue test, a large amount of energy will be accumulated at the defect or stress concentration area. As the crack propagates, the energy is gradually released, and the release of energy can lead to differences in the specific morphology, manifested as the length and deflection of the torn edge. In Figure 4a, the torn edge is relatively regular, indicating that the introduction of defects greatly reduces the difficulty of crack propagation. In Figure 5a, the torn edge is not perpendicular to the end face, indicating that the fatigue failure of the intact area is more difficult. Several rivers and stepped fatigue strips were observed in the extension area; however, the height undulation with defects was significantly higher than that at the weld toe. Furthermore, the fatigue strip with defects was small, and that at the weld toe was thick. This suggests that the presence or absence of defects in the early stages still affects the subsequent propagation of cracks. Moreover, regardless of a breakage at the defect or weld toe, several dimples were observed in the transient region, indicating that they were ductile fractures.

### 2.3. Comparison of Stress Concentration at Defect and Weld Toe

The actual specimen size was modeled in ABAQUS 2022. For specimens with defects, crack insertion can be achieved using the XFEM module provided by ABAQUS. The boundary conditions were set as one end fixed and the other end loaded with 100 Mpa tension. Hexahedral elements were used for meshing, and the specific type was C3D8R. Based on the results, the stress concentration factor was calculated, and the degree of stress concentration was determined by the stress concentration factor. The stress concentration factor is the ratio of the maximum stress to the average stress of the cross-section where it occurs [17].

To eliminate the influence of defects on the stress concentration at the weld toes, we established a defect-free model, and the calculation result is shown in Figure 6. Herein, the stress concentration coefficient at the weld toes was 2.246. The calculation results of samples containing 1 and 2.4 mm defects are shown in Figure 7 and Figure 8, and the stress concentration coefficients at the defects were 4.441 and 6.684, respectively; that is an increase of 98 and 198% compared with the non-defective samples.

Thus, the specimen breaks easily at weld toes without defects, as indicated by the simulation results. When a defect is introduced, the stress concentration at the defect is more severe than that at weld toes, which indicates that the specimen is more likely to break at the defect. Although there is a large difference in the stress concentration coefficient values with and without defects, in the actual situation, micro cracks, undercuts, and other defects can easily appear at the weld toes. Consequently, the existence of these defects is equivalent to tiny defects. Simultaneously, the probability of breaking at the weld toes is greatly increased, and that at the defect is relatively reduced.

## 3. Conclusions

(1)The fatigue limits of the specimens containing 1 and 2.4 mm defects were approximately 56.9 and 53.7 Mpa, respectively, which indicate a reduction of 19.4% and 23.9%, respectively, compared with complete welded specimens. Defects have a significant impact on the fatigue performance of the sample. However, even with a 2.4 mm defect, the sample still has about 70% of the load-bearing capacity of a complete sample. Allowing for 20% incomplete penetration is reasonable.(2)Most of the specimens broke at the defect. The crack source area broken at the defect was relatively regular, and that broken at the weld toe was in a curved state. Moreover, all of them were observed to have ductile fractures.(3)The stress concentration coefficients of no, 1 mm, and 2.4 mm defects were 2.246, 4.441, and 6.684, respectively; the stress concentration coefficients of 1 and 2.4 mm defects were increased by 98% and 198%, respectively, which significantly reduced the fatigue life of cross-welded joints.

## Figures and Tables

**Figure 1 materials-16-04751-f001:**
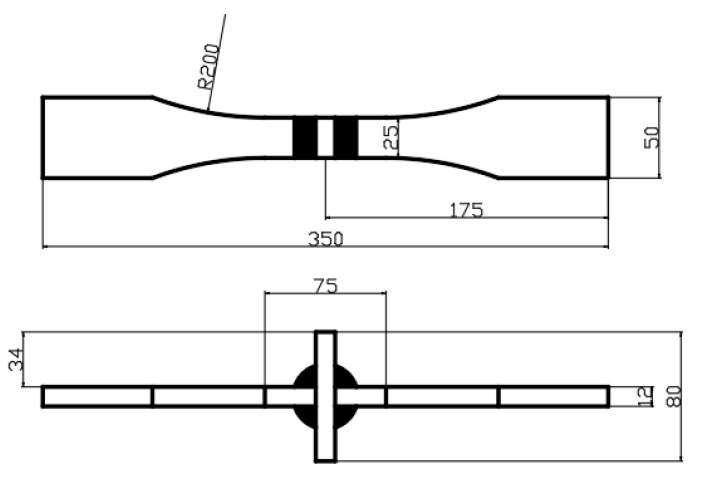
Original fatigue specimen.

**Figure 2 materials-16-04751-f002:**
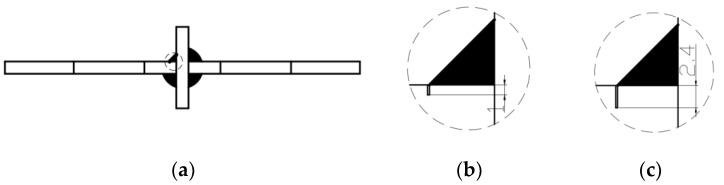
Fatigue specimen with defects of 1 mm and 2.4 mm. (**a**) Specimen size. (**b**) Local amplification of 1 mm defects. (**c**) Local amplification of 2.4 mm defects.

**Figure 3 materials-16-04751-f003:**
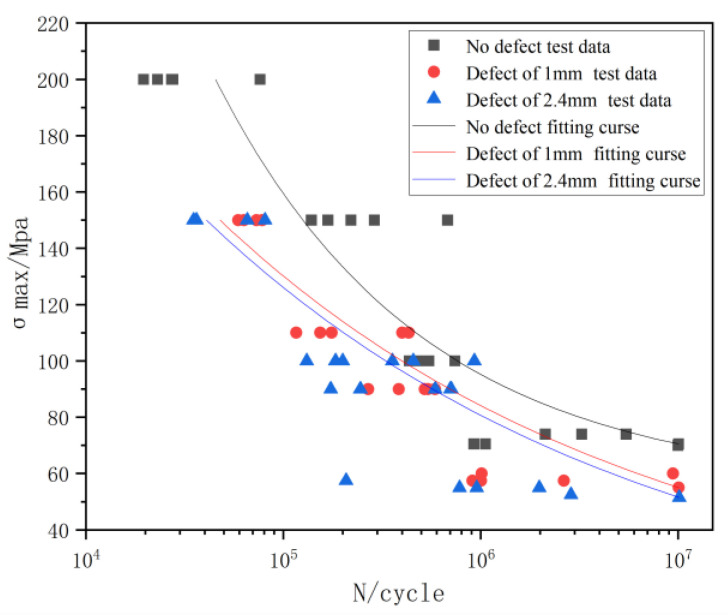
Comparison of S-N curves of defect-free and different defect sizes.

**Figure 4 materials-16-04751-f004:**
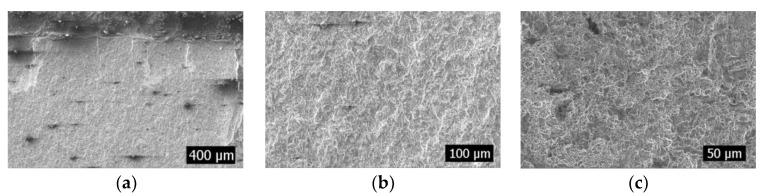
Fracture diagram at defect. (**a**): Crack source area; (**b**): Expansion zone; (**c**): Transient region.

**Figure 5 materials-16-04751-f005:**
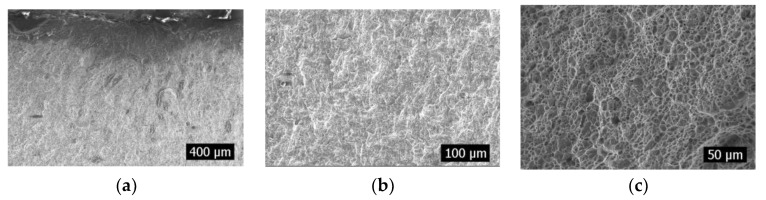
Fracture diagram at the weld toes. (**a**): Crack source area; (**b**): Expansion zone; (**c**): Transient region.

**Figure 6 materials-16-04751-f006:**
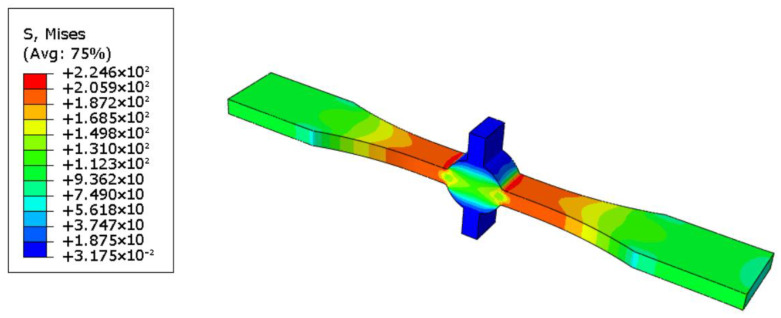
Cloud image of defect-free specimen.

**Figure 7 materials-16-04751-f007:**
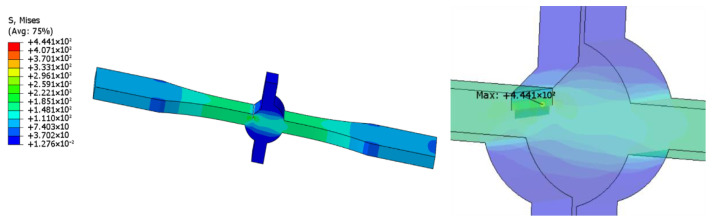
Cloud image of specimen with 1 mm defect.

**Figure 8 materials-16-04751-f008:**
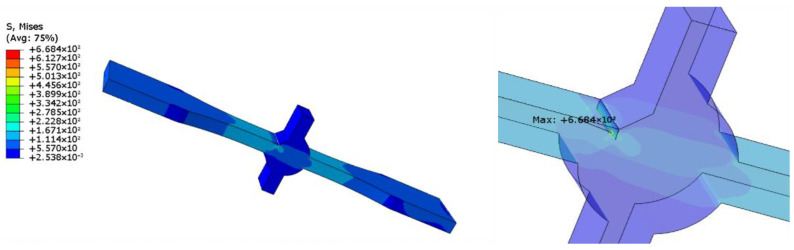
Cloud image of specimen with 2.4 mm defect.

**Table 1 materials-16-04751-t001:** Chemical composition of SMA490BW steel (wt%) [4].

C	Si	Mn	P	S	Cu	Cr	Ni
≤0.16	≤0.5	0.50~1.50	≤0.03	≤0.03	0.20~0.55	0.40~0.80	≤0.65

**Table 2 materials-16-04751-t002:** Mechanical properties of SMA490BW steel [4].

Yield Strength Rel/Mpa	Elastic Modulus E/GPa	Tensile Strength Rm/Mpa	Elongation Ae (%)	Tortuosity /(°)	Impact Energy KV/J
365	206	490~610	≥15	180 qualified	281.67

**Table 3 materials-16-04751-t003:** Results of SMA490BW steel cruciform joints with no defect (R = −1).

Defect Size	σ_max_/Mpa	*N*/Kilocycle	*S*	*N*_50_/Kilocycle
0	200	27.6, (76.3), 19.6, 23.1, 27.2	0.232	30.393
	150	(681.2), (289.8), 220.4, 168.5, 138.8	0.270	252.066
	100	(547.7), 489.2, (434.5), 740.6	0.099	541.877
	74	(3252.9), 5461.9, 2126.5	10^7^ fatigue limit Expressed in σ_max_ (maximum stress) as 70.9 Mpa	
	70.5	925.6, (1058.8), >10^4^, >10^4^, >10^4^		
	67	>10^4^, >10^4^		

**Table 4 materials-16-04751-t004:** Results of SMA490BW steel cruciform joint with defects of 1 mm (R = −1).

Defect Size	σ_max_/Mpa	*N*/Kilocycle	*S*	*N*_50_/Kilocycle
1 mm	150	63.2, 73.3, (59.2), (78.3)	0.056	11.784
	110	176.1, 153.9, (116.4), (400.7), (432.9)	0.257	222.655
	90	(269.9), (385.6), (522.7), (541.2), (587.5)	0.139	444.213
	60	(1011.7), 9452.5	10^7^ fatigue limit Expressed in σ_max_ (maximum stress) as 57.2 Mpa	
	57.5	1006.0, (911.5), (2643.4), >10^4^		
	55	>10^4^, >10^4^, >10^4^		

**Table 5 materials-16-04751-t005:** Results of SMA490BW steel cruciform joint with defects of 2.4 mm (R = −1).

Defect Size	σ_max_/Mpa	*N*/Kilocycle	*S*	*N*_50_/Kilocycle
2.4 mm	150	35.1, 36.4, (81.0), (65.7)	0.183	51.064
	100	200.3, 131.7, 184.2, (357.5), (455.4), (929.2)	0.079	300.416
	90	174.3, 245.8, (703.7), (710.6), (588.6)	0.285	417.006
	57.5	208.1	10^7^ fatigue limit Expressed in σ_max_ (maximum stress) as 53.7 Mpa	
	55	1983.5, 955.7, 782.6, >10^4^		
	52.5	2874.0, >10^4^, >10^4^, >10^4^		
	50	>10^4^		

**Table 6 materials-16-04751-t006:** S-N curve-fitting equation after defect treatment of SMA490BW steel butt joint.

Defect Size	S-N Curve-Fitting Equation	Scope of Application
0	$\mathrm{lg}N=9.865-2.410\times \mathrm{lg}(\sigma_{\max}-55.1)$	10^4^~10^7^
1 mm	$\mathrm{lg}N=15.48-5\times \mathrm{lg}(\sigma_{\max}-5.45)$	10^4^~10^7^
2.4 mm	$\mathrm{lg}N=15.84-5.16\times \mathrm{lg}\sigma_{\max}$	10^4^~10^7^

## Data Availability

Restrictions apply to the availability of these data. Data are available from the corresponding author with the permission of Shenyang Aerospace University.

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
