# Peer review of "Effect of Welding Defects on Fatigue Properties of SWA490BW Steel Cruciform Welded Joints"

_materials, 2023, doi:10.3390/ma16134751_

Round 1

Reviewer 1 Report

The article under review is entitled: Effect of welding defects on fatigue properties of SWA490BW steel cruciform welded joints.

The basic structure of the work itself is correct, i.e. the introduction, the object of research, the results and conclusions. Unfortunately, going into the content of the publication in detail, you can come across numerous shortcomings.

In the introduction, 23 scientific publications were cited. Unfortunately, in many cases, several were uploaded at once, without proper citation of the content and novelties contained in them (this should be expanded).

Later, the research facility, methodology and results were presented. Unfortunately, there are also shortcomings here.

Was the shape of the sample based on a standard?

Where did the chemical composition of materials come from?

What is the source of the strength properties?

Units were misspelled (Mpa was used instead of MPa).

lack of information on boundary conditions and finite element mesh used for numerical analysis

no information about the software version.

What is the cloud image presented in Figures 8 and 9?

No detailed analysis of fractographic images.

Poor results - need to be expanded.

In general, the publication should be expanded and developed before being published in the journal Metals.

There are single linguistic and lexical errors in the article.

Author Response

  1. Was the shape of the sample based on a standard?

The overall fatigue test piece of the welded joint is carried out in accordance with the "HB5287-1996 Metal Material Axial Loading Fatigue Test Method".

  1. Where did the chemical composition of materials come from?

The composition of the material comes from reference [4].

  1. What is the source of the strength properties?

The composition of the material comes from reference [4].

  1. Units were misspelled (Mpa was used instead of MPa).

Units have been revised

  1. lack of information on boundary conditions and finite element mesh used for numerical analysis

The boundary conditions of the specimen are fixed at one end and subjected to a tensile load at the other end. The mesh is based on the C3D8R elements under the hexahedral family.

  1. no information about the software version.

The software version used is ABAQUS 2022.

  1. What is the cloud image presented in Figures 8 and 9?

The cloud images presented in Figures 8 and 9 show the stress distribution of the specimen after being subjected to a load of 100 Mpa, with defects of 1 mm and 2.4 mm, respectively.

  1. No detailed analysis of fractographic images.

The information for the fracture image has already been translated into English.

  1. Poor results - need to be expanded.

The results have already been supplemented in the text.

Reviewer 2 Report

The paper shows interesting research results. There are no comments on the data presented in the article. But I would like to clarify whether there are corrosion damages on these welded joints and whether they contribute to the destruction of the elements? This should be indicated in the Introduction when setting the purpose of the study, so that it is clear why corrosion tests are not carried out.

There is also a note on the design of the manuscript structure: sections 3 and 4 are not highlighted, which should contain the results and discussion of the data.

1. The study is devoted to the influence of hidden defects on the strength characteristics of welded joints.

2. The topic is relevant, but the presented results do not provide information about the service life of the studied products.

3. The results of the study provide an understanding of the degree of influence of hidden defects on the properties of welded joints, but they cannot be called new or exclusive.

4. In the section "Materials and Methods" it is necessary to highlight the description of the methods used in the study. And the results and discussion need to be moved to separate sections. The authors should consider the effect of corrosion damage on these welded joints. It is also necessary to establish specific service life of the studied products.

5. The conclusions correspond to the results obtained. It should also indicate the further direction of research and the application of the results obtained.

6. The references are appropriate.

7. Figures 2 and 3 can be combined by converting 3-b into 2-c.

Author Response

    1. The study is devoted to the influence of hidden defects on the strength characteristics of welded joints.

    The influence has been given in the article’s introduction.

    1. The topic is relevant, but the presented results do not provide information about the service life of the studied products.

    The introduction of the article states that the service life of the steering mechanism shall not be less than 3 million kilometers.

    1. The results of the study provide an understanding of the degree of influence of hidden defects on the properties of welded joints, but they cannot be called new or exclusive.

    The description has been modified.

    1. In the section "Materials and Methods" it is necessary to highlight the description of the methods used in the study. And the results and discussion need to be moved to separate sections. The authors should consider the effect of corrosion damage on these welded joints. It is also necessary to establish specific service life of the studied products.

    1. The conclusions correspond to the results obtained. It should also indicate the further direction of research and the application of the results obtained.

    1. The references are appropriate.

    1. Figures 2 and 3 can be combined by converting 3-b into 2-c.

    Image has been modified.